# Dyadic Learning
# in Recurrent and Feedforward Models

**Rasmus Kjær Høier** [*]     **Kirill Kalinin**[†]     **Maxence Ernoult**[‡]     **Christopher Zach**[§]

## Abstract

From electrical to biological circuits, feedback plays a critical role in amplifying, dampening and stabilizing signals. In local activity difference based alternatives to backpropagation, feedback connections are used to propagate learning signals in deep neural networks. We propose a saddle-point based framework using dyadic (two-state) neurons for training a family of parameterized models, which include the symmetric Hopfield model, pure feedforward networks and a less explored *skew-symmetric Hopfield variant*. The resulting learning method reduces to equilibrium propagation (EP) for symmetric Hopfield models and to dual propagation (DP) for feedforward networks, while the skew-symmetric Hopfield setting yields a new method with desirable robustness properties. Experimentally we demonstrate that the new skew-symmetric Hopfield model performs on par with EP and DP in terms of the resulting model predictive performance, while exhibiting enhanced robustness to input changes and strong feedback and is less inclined to neural saturation. We identify the fundamentally different types of feedback signals propagated in each model as the main cause of differences in robustness and saturation.

## 1 Introduction

It has been proposed that the brain may perform some instance of approximate backpropagation, using neural activity differences to represent error signals [1, 2]. Activity difference based approaches come in two main flavours: (i) using *temporal* differences between activities at different times to represent error signals [3, 4, 5], or (ii) using *spatial* differences by subtracting activities of distinct neurons or neuronal compartments [6, 7, 8]. Temporal activity difference methods require at least two relaxations to equilibrium, or infinitely many through slow adiabatic oscillations of the *same* circuit [9], yet at the expense of latency. Conversely, spatial activity difference methods require neurons to multiplex by propagating activities and finite difference error signals simultaneously and some degree of weight sharing, which may be problematic from a biological perspective.

In this manuscript, we take a closer look at the temporal activity difference based algorithm equilibrium propagation (EP) [4] for symmetric Hopfield models (HM), the spatial activity difference based algorithm dual propagation (DP) for feedforward models (FFM) [8], and the fundamental role feedback plays in each of the algorithms. In the context of this work, feedback refers to both the error signal propagated in a supervised learning setting and to top-down connections in a layered network architecture (and therefore introducing cycles in the underlying computational graph). In DP, the feedback signals reduce to top-down (finite difference) errors due to the lack of recurrent connections in feedforward models. While this is an asset for inference speed in practice, recurrent connections are a key feature underpinning attention and robustness in biological systems. On the

---

[*]Chalmers University of Technology. Work begun during internship at Microsoft Research.
[†]Microsoft Research
[‡]RAIN AI
[§]Chalmers University of Technology

Second Workshop on Machine Learning with New Compute Paradigms at NeurIPS 2024(MLNCP 2024).

other hand, in EP (and Hopfield nets in general) *positive* feedback is always present: the sign of the signal send from neuron A to neuron B always matches the sign of the signal sent from B to A. This is a consequence of the symmetry of the underlying Hopfield model ($W_{ij} = W_{ji}$). Positive feedback connections in symmetric Hopfield nets yield many desirable properties, but may also lead to excessive amplification of neural activities. This is not too surprising as high sensitivity to input changes and excessive saturation are known features of positive feedback [10].

As FFMs without an explicit error signal lack any kind of feedback from downstream layers and HMs suffer from the aforementioned limitations of persisting positive feedback, it is natural to ask what a *negative* feedback model would look and behave like? As a symmetric Hopfield model results in positive feedback, a *skew-symmetric Hopfield model* (SSHM), with with $W_{ij} = -W_{ji}$, would instead provide negative feedback. Negative feedback loops are typically used to decrease sensitivity to input perturbations and increase the linearity of a system [10] which could respectively endow the resulting model with a degree of biological plausibility and alleviate vanishing gradient issues during training.

We establish a family of models and their associated learning dynamics, which recover as special cases symmetric Hopfield models (HM), feedforward models (FFM) and skew-symmetric Hopfield models (SSHM). In the HM setting, the neural state and synaptic weight dynamics correspond to EP, and the FFM setting results in DP. The SSHM scenario yields a two-phased algorithm which exhibits reduced sensitivity to input perturbations and also to decreased neural saturation.

While the role of feedback in the brain is much more diverse and subject to constraints we are not modelling (such as weight transport and, Dale's principle) we hope this study helps bring attention to the importance of negative feedback in the context of learning. The importance of negative feedback is well known in both neuroscience and control theory, yet in the context of biologically inspired learning algorithms negative feedback has received little to no attention.

## 2    Related work

**Contrastive learning algorithms.** In energy-based models, inference amounts to energy minimization. Training can in this setting be achieved by contrasting the states inferred when applying different perturbations to the output units. In variations of contrastive Hebbian learning (CHL) [3, 11, 5], this is done by clamping output units. In EP, the output units are instead weakly nudged by a teaching signal derived from a loss function. EP and CHL typically assume a symmetric connectivity, which for instance naturally map to resistive networks where top-down currents inherently flow through the weights transpose [12, 13, 14, 15].

**Credit assignment with dual compartments.** DP introduces "dyadic" neurons, which permits representing errors and activities simultaneously as the difference and mean of each neurons compartments, which in terms permits turning the two distinct inference phases of CHL or EP into a single phase. The underlying motivation is conceptually similar, albeit different in terms of resulting algorithms, to the neuroscience inspired approximations of backpropagation [16, 7] which achieve single-phased learning by modelling pyramidal neurons with distinct compartments for integrating top-down and bottom-up signals.

**Lifted neural networks.** Lifted neural networks have roots in the community of mathematical optimization and frame neural network training as bilevel optimization over weights and activations [17, 18, 19, 20, 21, 22]. Although predating the term *lifted network* [23] also fits in this category. As in variants of CHL, neurons in lifted neural networks rely only on local information and require less synchronization than backpropagation. In the context of lifted networks robustness to input perturbations has previously been explored in [24].

## 3    A Saddle point objective for dyadic learning

We let subscript $k$ denote row index and subscript $\setminus k$ denote the removal of the $k'$th element from a vector. Starting from the optimization problem

$$\min_{W} \ell(s, y) \quad \text{s.t. } \forall \, \mathrm{k} \; s_k = \arg\min_{s_k'} E_k(s_k', s_{\setminus k}, W_{k, \setminus k}, \theta_0) \tag{1}$$

we derive the following relaxation (see details in appendix A).

$$\min_W \min_{s^+} \max_{s^-} \tfrac{1}{2}\ell(s^+,y) + \tfrac{1}{2}\ell(s^-,y) + \tfrac{1}{\beta}\sum_k E_k(s_k^+,\bar{s}_{\backslash k}) - E_k(s_k^-,\bar{s}_{\backslash k}) \tag{2}$$

In the case of a Hopfield-like $E_k$ this correponds to performing gradient descent on the following objective with respect to the weights.

$$\mathcal{L}(\theta_0, W) = \min_{s^+}\max_{s^-} \tfrac{\beta}{2}\ell(s^+,y) + \tfrac{\beta}{2}\ell(s^-,y) \tag{3}$$

$$+ G(s^+) - G(s^-) - (s^+ - s^-)^\top \theta_0 x - \tfrac{1}{4}\begin{pmatrix} s^+ \\ s^- \end{pmatrix}^\top \underbrace{\begin{pmatrix} W+W^\top & W-W^\top \\ -W+W^\top & -W-W^\top \end{pmatrix}}_{:=\widetilde{W}} \begin{pmatrix} s^+ \\ s^- \end{pmatrix}$$

Here $\theta_0$ is an projection mapping datapoints $x$ to the space of neural activities $s$, while $W$ governs interactions between elements of $s$. The structure of $W$ has profound impact on how this optimization can be physically realized. The general case of an entirely unstructured weight matrix $W$ requires neurons to perform substantial multiplexing, making it challenging for physical implementations. Choosing instead to parametrize $W_\lambda(\theta) := \theta + (2\lambda - 1)\theta^\top$, where $\theta$ is lower triangular, allows smoothly interpolating from symmetric Hopfield models, through feedforward models, to skew-symmetric Hopfield models (each of which offer certain benefits for physical implementations). Employing this parametrization yields the following objective.

$$\mathcal{L}_\lambda(\theta_0,\theta) = \min_{s^+}\max_{s^-} \tfrac{\beta}{2}\ell(s^+,y) + \tfrac{\beta}{2}\ell(s^-,y) \tag{4}$$

$$+ G(s^+) - G(s^-) - (s^+ - s^-)^\top \theta_0 x - \tfrac{1}{4}\begin{pmatrix} s^+ \\ s^- \end{pmatrix}^\top \underbrace{\begin{pmatrix} \lambda(\theta+\theta^\top) & \bar{\lambda}(\theta-\theta^\top) \\ \bar{\lambda}(-\theta+\theta^\top) & \lambda(-\theta-\theta^\top) \end{pmatrix}}_{:=\widetilde{W}_\lambda} \begin{pmatrix} s^+ \\ s^- \end{pmatrix}$$

In the following we will restrict our focus to the more hardware friendly settings offered by $\mathcal{L}_\lambda$.

**Dynamics.**  Training amounts to solving $\min_\theta \mathcal{L}_\lambda(\theta)$. Let the accent $*$ denote optimality, then the gradient of $\mathcal{L}_\lambda(\theta)$ with respect to $\theta$ is

$$\tfrac{1}{\beta}\nabla_\theta \mathcal{L}_\lambda(\theta) = -\tfrac{\lambda}{2\beta}(\overset{*}{s}^+\overset{*}{s}^{+\top} - \overset{*}{s}^-\overset{*}{s}^{-\top}) + \tfrac{\bar{\lambda}}{2\beta}(\overset{*}{s}^-\overset{*}{s}^{+\top} - \overset{*}{s}^+\overset{*}{s}^{-\top}) \tag{5}$$

As $\widetilde{W}_\lambda$ is already symmetric, the gradient with respect to $s^\pm$ is simple to compute

$$\begin{pmatrix} \nabla_{s^+}\mathcal{L}_\lambda \\ \nabla_{s^-}\mathcal{L}_\lambda \end{pmatrix} = \begin{pmatrix} \nabla_{s^+}\ell(s^+) \\ \nabla_{s^-}\ell(s^-) \end{pmatrix} + \tfrac{1}{\beta}\left(\begin{pmatrix} f^{-1}(s^+) \\ -f^{-1}(s^-) \end{pmatrix} - \widetilde{W}_\lambda \begin{pmatrix} s^+ \\ s^- \end{pmatrix} - \theta_0 \begin{pmatrix} x \\ -x \end{pmatrix}\right). \tag{6}$$

and the stationary conditions for $\overset{*}{s}^\pm$ are

$$\overset{*}{s}^\pm = f\left(\theta_0 x + \tfrac{1}{2}W_\lambda(\overset{*}{s}^+ + \overset{*}{s}^-) \pm \tfrac{1}{2}W_\lambda^\top(\overset{*}{s}^+ - \overset{*}{s}^-) \pm \beta\nabla\ell(\overset{*}{s}^\pm)\right). \tag{7}$$

We determine $\overset{*}{s}^\pm$ using dampened fixed-point iterations (see appendix B for details) of the preactivations $a^\pm$ (defined through the relation $s^\pm = f(a^\pm)$) of the form

$$a^\pm \leftarrow (1-\eta)a^\pm + \eta\left(\theta(\lambda s^\pm + \bar{\lambda}s^\mp) + \theta^\top(\lambda s^\pm - \bar{\lambda}s^\mp) + \theta_0 x \mp \beta\nabla\ell(s^\pm)\right). \tag{8}$$

In the remaining part of the paper we will further restrict $\theta$ to be strictly lower block triangular (rather than just strictly lower triangular). We write this in terms of layerwise weight matrices $\theta_k$.

$$\theta = \begin{pmatrix} 0 & 0 & \dots & 0 \\ \theta_1 & 0 & \dots & 0 \\ 0 & \theta_2 & \dots & 0 \\ \vdots & \vdots & \ddots & 0 \end{pmatrix} \tag{9}$$

This approach is commonly applied to obtain layered Hopfield networks. In this setting we get mirror descent dynamics for the hidden layers of the form

$$a_k^\pm \leftarrow (1-\eta)a_k^\pm + \eta\left(\theta_{k-1}(\lambda s_{k-1}^\pm + \bar{\lambda}s_{k-1}^\mp) + \theta_k^\top(\lambda s_{k+1}^\pm - \bar{\lambda}s_{k+1}^\mp)\right), \tag{10}$$

where the subscript $k$ now denotes layer index. When $\eta = 1$ this reduces to layer-wise closed form updates akin to those employed in EP [25].

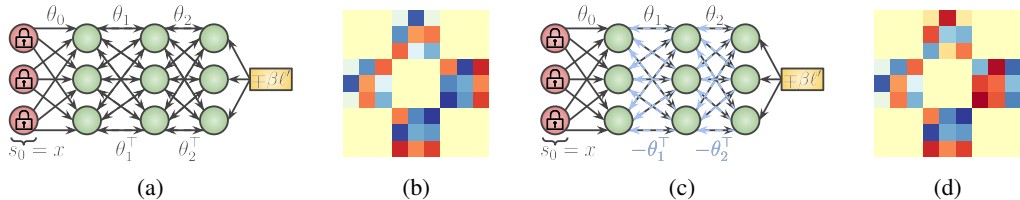

<p style="text-align:center">(a)          (b)          (c)          (d)</p>

Figure 1: Sketch **1a** and adjacency matrix **1b** of a symmetric Hopfield model. Sketch **1c** and adjacency matrix **1d** of a skew-symmetric Hopfield model. Dashed blue arrows denotes opposite sign of the corresponding forward connection. Note that the input projection $\theta_0$ is not part of the adjacency matrix $W_\lambda$.

**Recovering symmetric Hopfield models trained with EP.** In the case $\lambda = 1$ and (i.e. $\bar{\lambda} = 0$) then the saddle-point problem equation 4 decouples into two separate minimization problems over $s^+$ and $s^-$, respectively. This corresponds to the two phases of EP in a deep Hopfield network, yielding the hidden layer dynamics:

$$a_k^\pm \leftarrow (1 - \eta)a_k^\pm + \eta \left( \theta_{k-1} s_{k-1}^\pm + \theta_k^\top s_{k+1}^\pm \right) \tag{11}$$

In a hardware realization, this means the two sub-problems can be either solved in parallel (which requires maintaining two identical sets of weights) or sequentially using the same hardware (which avoids weight doubling, but increases runtime). Figure 1a and 1b illustrates the connectivity matrix $\widetilde{W}_{\lambda=1}$ for a small network.

**Recovering feedforward models trained with DP.** The choice $\lambda = \bar{\lambda} = 1/2$ recovers the DP dynamics.

$$a_k^\pm \leftarrow (1 - \eta)a_k^\pm + \eta \left( \tfrac{1}{2}\theta_{k-1}(s_{k-1}^\pm + s_{k-1}^\mp) + \tfrac{1}{2}\theta_k^\top (s_{k+1}^\pm - \bar{s}_{k+1}^\mp) \right) \tag{12}$$

In the absence of a teaching signal ($\beta \to 0$) then $s_k^+ = s_k^-$ for all $k$, hence the feedback term $\tfrac{1}{2}\theta_k^\top (s_{k+1}^\pm - \bar{s}_{k+1}^\mp)$ vanishes and the model behaves exactly like a feedforward model. For hardware implementations it is noteworthy that the two compartments $s_{k,i}^+$ and $s_{k,i}^-$ of a neuron $i$ in layer $k$ receive the same bottom up and the same absolute feedback signal (but opposite sign), which allows them to share weights in physical realizations.

**Skew-symmetric Hopfield models** Fixing $\lambda = 0$ (i.e. $\bar{\lambda} = 1$) implies that $W_{\lambda=0}$ is skew-symmetric (see Fig. 1(c–d)), and only the blocks governing interactions between $s^+$ and $s^-$ remain in $\widetilde{W}_\lambda$. The dynamics in the layered setting we consider here are

$$a_k^\pm \leftarrow (1 - \eta)a_k^\pm + \eta \left( \theta_{k-1} s_{k-1}^\mp - \theta_k^\top s_{k+1}^\mp \right) \tag{13}$$

A critical observation here is that states with odd layer indices of $a^\pm$ only interact directly with $a^\mp$ states with even indices, and vice versa. This means that the global minmax problem over $a^+$ and $a^-$ decouples into two independent minmax problems: one minmax problem over odd layers of $a^+$ and even layers of $a^-$, and another over even layers of $a^+$ and odd layers of $a^-$. Consequently, as in the case of the symmetric Hopfield model this setting has two phases, but unlike the symmetric setting each phase here corresponds to a minmax rather than a pure minimization problem.

In early research on *binary* skew-symmetric Hopfield networks it was shown that skew-symmetric weights lead to cycles in binary Hopfield models [26, 27]. We do observe oscillating behaviour in our continuous SSHM when applying analogous closed form updates, but not when employing a small inference stepsize (we used $\eta = 0.05$) in the dampened fixed point iterations. A similar skew-symmetric connectivity has previously been explored in [28] to preserve long term dependencies and avoid vanishing gradients in the context of sequence data (this is different from our setting where input data is static).

**Lipschitz properties** The skew-symmetric Hopfield model gives rise to remarkably different Lipschitz properties. Since a fixed point satisfies $s = f(W_\lambda s + x)$, we have via implicit differentiation

$$\frac{ds}{dx} = f'(W_\lambda s + x)\left( W_\lambda \frac{ds}{dx} + \mathtt{I} \right). \tag{14}$$

<p style="text-align:center">4</p>

Table 1: Test accuracy and Lipschitz constants (of hidden layers and the full network).

| | MNIST | | | FashionMNIST | | |
|---|---|---|---|---|---|---|
| | Test acc (%) | $L_{\text{hidden}}$ | $L_{\text{output}}$ | Test acc (%) | $L_{\text{hidden}}$ | $L_{\text{output}}$ |
| SSHM | $98.46 \pm 0.04$ | $1.0\pm$ 7e-7 | $0.41 \pm 0.05$ | $89.45 \pm 0.07$ | $1.0\pm$ 3e-7 | $0.53 \pm 0.03$ |
| FFM | $98.37 \pm 0.09$ | $116.4 \pm 12.9$ | $95.75 \pm 7.44$ | $89.11 \pm 0.17$ | $141.6 \pm 43.6$ | $138.17 \pm 43.12$ |
| HM | $97.97 \pm 0.12$ | $931.0 \pm 492.0$ | $24.07 \pm 10.41$ | $89.16 \pm 0.25$ | $1257.1 \pm 576.7$ | $31.83 \pm 16.88$ |

By *assuming linear activation mappings* for the moment, we obtain $\frac{ds}{dx} = (\mathtt{I} - W_\lambda)^{-1}$. In the skew-symmetric setting ($\lambda = 0$), $W_0$ has a decomposition of the form $W_0 = Q\Sigma Q^\top$ where $Q$ is orthogonal and $\Sigma$ is a matrix with only zeros and $2 \times 2$ blocks of the form $\left(\begin{smallmatrix} 0 & \lambda_i \\ -\lambda_i & 0 \end{smallmatrix}\right)$ on its diagonal. Consequently,

$$W_0^\top W_0 = Q\Sigma^\top \Sigma Q^\top = Q \operatorname{diag}(\lambda_i^2)_i Q^\top, \tag{15}$$

and therefore $W_0^\top W_0$ has only non-negative eigenvalues. Hence, $(\mathtt{I}+W_0)^\top (\mathtt{I}+W_0) = \mathtt{I}+W_0^\top + W_0 + W_0^\top W_0 = \mathtt{I} + W_0^\top W_0$ has only eigenvalues $\geq 1$ and therefore $(\mathtt{I} + W_0^\top W_0)^{-1}$ has all eigenvalues $\in [0, 1]$. Finally,

$$\|(\mathtt{I} + W_0)^{-1}\|_2 = \sqrt{\lambda_{\max}((\mathtt{I} + W_0^\top W_0)^{-1})} \in [0, 1] \tag{16}$$

and the mapping $x \mapsto (\mathtt{I}+W_0)^{-1}x$ is 1-Lipschitz. In Appendix C we show that this property carries over to element-wise non-expansive activation functions. This is exciting as it means the network has a degree of built-in robustness to adversarial perturbations [29].

## 4 Experiments

We train MLPs with 4 hidden layers with 500 neurons each, using the three special cases $\lambda = 1$ (HM), $\lambda = 1/2$ (FFM) and $\lambda = 0$ (SSHM). The models were trained on MNIST and FashionM-NIST. EP (which our learning dynamics reduces to for $\lambda = 1$) typically works best with bounded activation functions so we trained all models with a tempered hard sigmoid activation function $\sigma(x) = \max(0, \min(1, \frac{x}{2}))$. This activation function was introduced in [30] to reduce the degree of saturation (neurons with value 0 or value 1). For reduced runtime we used undamped fixed-point iterations when possible (FFM and HM) and dampened one otherwise (SSHM) in order to ensure stable inference.

**Weak nudging.** We carry out experiments applying weak centered nudging (analogous to the centered nudging introduced in [30]). As shown in table 1 each of the models manage to solve both the MNIST and Fashion MNIST tasks, although the HM model lacks a bit behind on MNIST. As predicted by the theory, the upper bound on the sensitivity (measured by the Lipschitz constant $L_{\text{hidden}}$) of the hidden units to changes in their input $y = \theta_0 x$ is smallest for the skew-symmetric Hopfield model. We also report the Lipschitz estimate $L_{\text{output}}$ for the output units as a function of the network input $x$, where $L_{\text{output}}$ is the spectral norm of

$$(\mathtt{I}\, 0 \cdots 0)(\mathtt{I} - W_\lambda)^{-1} \begin{pmatrix} \vdots \\ 0 \\ \theta_0 \end{pmatrix}. \tag{17}$$

The symmetric Hopfield model achieves a smaller $L_{\text{output}}$ than the feedforward model, which is in line with a recent study finding EP to be more adversarially robust than feedforward models [31]. Figure 2b and 2e show that the SSHM model consistently generalizes better (difference between validation and training performance is smaller). Figure 2c and 2f show that the fraction of saturated neurons is highest for the HM model and lowest for the SSHM model.

**Strong nudging.** Although EP and DP to a large extent are motivated as algorithms for new non-von Neumann compute platforms they require $\beta$ to be small. This is problematic as a small nudging signal $\pm\beta\ell'$ can drown out in noisy circuits. We carry out an experiment where all settings are as in the weak feedback experiments except that for each epoch we increment $\beta$ by 0.05. As expected the use of positive feedback in the symmetric Hopfield model leads to rapid performance degradation. The use of negative feedback in the skew-symmetric Hopfield model permits stable learning with significantly stronger nudging strength.

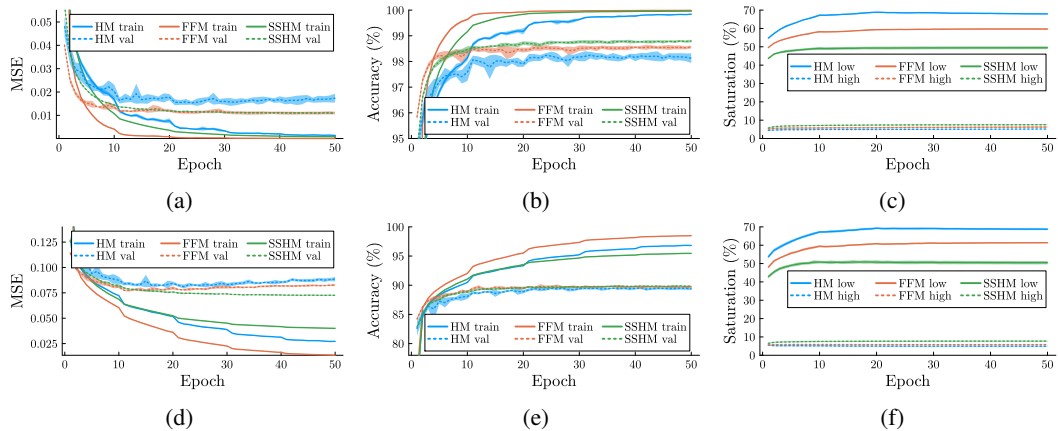

Figure 2: MNIST (top) and FashionMNIST (bottom) loss (1st column), accuracy (middle) and fraction of saturated neural activations (right) for a 500-500-500-500-10 hard sigmoid network. Shaded areas denote one standard deviation.

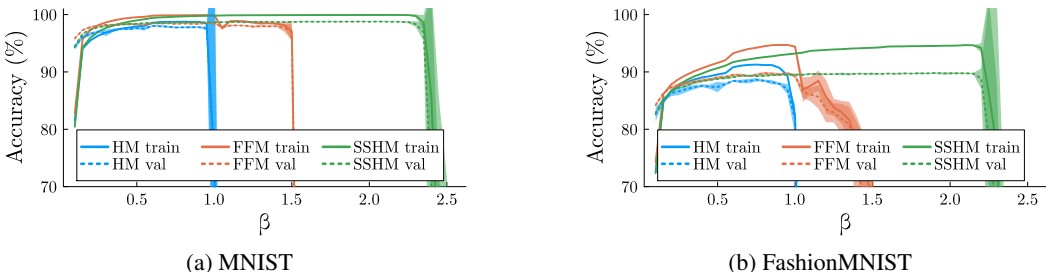

Figure 3: Impact of gradually increasing $\beta$ by $0.05$ per epoch for MNIST and FashionMNIST.

## 5 Discussion

The proposed saddle-point framework in Eq. 4 encompasses training of a spectrum of algorithms parameterized in terms of $\lambda$. From a physical implementation perspective, only the Hopfield model with symmetric, skew-symmetric, and strictly lower triangular (feedforward model) connectivity matrices are truly appealing as they avoid excessive weight copying and multiplexing. Conveniently, such models can be implemented in analog compute platforms with programmable weights. For example, spatial light modulators allow one to program weights with up to 10 bit precision [32]. With such a device, the matrix-vector multiplication could be efficiently computed in the optical domain using light, while the required nonlinearities and other operations could be further realized in the analog electronic domain, resulting in the efficient opto-electronic Hopfield networks. In practice, real physical systems suffer from various nonidealities, which may result in approximately symmetric or skew-symmetric matrices. The impact of such physical imperfections on the dynamical properties of Hopfield-like networks is an exciting direction to investigate.

Due to the negative feedback connections the skew-symmetric Hopfield model exhibits a strong inherent degree of robustness to perturbations, and is able to learn in the presence of a strong teaching signal. These are aspects we expect to be important in noisy substrates such as the brain as well as neuromorphic hardware. We hope this work will encourage further research into the role of negative feedback in learning.

**Outlook.** Due to hardware concerns we have not explored the general case $\lambda \notin \{0, 1/2, 1\}$. One use for $\lambda \notin \{0, 1/2, 1\}$ would be to start from a pre-trained feedforward model and slowly alter $\lambda$ while fine-tuning the weights. This would allow a kind of translating between between feedforward ($\lambda = 1/2$), Hopfield models ($\lambda = 1$) and skew-symmetric Hopfield models ($\lambda = 0$). It is also possible to choose $\lambda$ on a layer by layer basis. This would be an alternative way to approach the joint training of chained feedforward and Hopfield models proposed in [33].

## Acknowledgments

This work was supported by the Wallenberg AI, Autonomous Systems and Software Program (WASP) funded by the Knut and Alice Wallenberg Foundation, and by the Chalmers AI Research Centre (CHAIR). The experiments were enabled by the supercomputing resource Berzelius provided by National Supercomputer Centre at Linköping University and the Knut and Alice Wallenberg foundation. Many thanks to Jannes Gladrow for valuable guidance and suggestions during the early stages of this project and to Jack Kendall and Benjamin Scellier for helpful discussions, regarding implementations of negative feedback in resistive networks.

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

## A Deriving the objective

Consider the following constrained optimization problem.

$$\min_{W} \ell(s, y) \quad \text{s.t. } \forall \text{ k } s_k = \arg\min_{s_k'} E_k(s_k', s_{\setminus k}, W_{k, \setminus k}) \tag{18}$$

We introduce the notation $s_k$ denoting the k'th neuron in the state vector $s$ and $s_{\setminus k}$ denoting the state vector $s$ with the k'th element removed. $W_{k, \setminus k}$ denotes the $k'th$ row of $W$ with the $k'th$ element removed, but for brevity we omit this argument in the following and simply write $E_k(s_k, s_{\setminus k})$. Furthermore we make the assumption that $E_k(s_k, s_{\setminus k})$ has a unique minimizer $s_k^*$. Next we define $\bar{s} := \frac{1}{2}(s^+ + s^-)$ then we can rewrite this as

$$\min_{W} \min_{s^+} \max_{s^-} \tfrac{1}{2}\ell(s^+, y) + \tfrac{1}{2}\ell(s^-, y) \quad \text{s.t. } \forall \text{ k } s_k^+ = \arg\min_{s_k'} E_k(s_k', \bar{s}_{\setminus k}) \tag{19}$$

$$s_k^- = \arg\min_{s_k'} E_k(s_k', \bar{s}_{\setminus k}).$$

Since we (at the moment) are strictly enforcing $s_k^+ = s_k^- = s_k^*$ via the constraints the additional minimization and maximization does not influence the objective and problem 19 is equivalent to problem 18. The constraints associated with $s_k^\pm$ can now be rewritten using the optimal value reformulation (OVR) [34] yielding

$$\min_{W} \min_{s^+} \max_{s^-} \tfrac{1}{2}\ell(s^+, y) + \tfrac{1}{2}\ell(s^-, y) \tag{20}$$

$$\text{s.t. } \forall \text{ k } \begin{cases} E_k(s_k^+, \bar{s}_{\setminus k}) \leq \min_{s_k'} E_k(s_k', \bar{s}_{\setminus k}) \\ E_k(s_k^-, \bar{s}_{\setminus k}) \leq \min_{s_k'} E_k(s_k', \bar{s}_{\setminus k}) \end{cases}$$

Rather than strictly enforcing these constraints we add penalty terms that enforce that approximately enforce the constraint weighted by a gain factor $1/\beta$, where $\beta > 0$. In the limit $\beta \to 0$ the constraints are satisfied. The constraints associated with the $s^-$ (a maximization) results in the penalty term

$$\tfrac{1}{\beta} \sum_k \min_{s_k'} E_k(s_k', \bar{s}_{\setminus k}) - E_k(s_k^-, \bar{s}_{\setminus k}) \tag{21}$$

and the constraints associated with the minimization over $s^+$ results in the penalty term

$$\tfrac{1}{\beta} \sum_k E_k(s_k^+, \bar{s}_{\setminus k}) - \min_{s_k'} E_k(s_k', \bar{s}_{\setminus k}). \tag{22}$$

After accounting for terms canceling out the inclusion of these penalty terms yield the following Relaxation of problem 18.

$$\min_{W} \min_{s^+} \max_{s^-} \tfrac{1}{2}\ell(s^+, y) + \tfrac{1}{2}\ell(s^-, y) + \tfrac{1}{\beta} \sum_k E_k(s_k^+, \bar{s}_{\setminus k}) - E_k(s_k^-, \bar{s}_{\setminus k}) \tag{23}$$

## A.1 Specializing to the Hopfield energy case

Next we consider the special case of a Hopfield-like energy. We explicitly define an input projection weight matrix $\theta_0$ that maps data $x$ to the space of the activities $s$. By $\theta_{0,k}$ we denote the $k'th$ row of $\theta_0$.

$$E_k(s_k, s_{\setminus k}) = G(s_k) - s_k W_{k \setminus k} s_{\setminus k} - s_k \theta_{0,k} x \tag{24}$$

*Remark* A.1. This formulation yields closed form updates of the form $s = f(Ws + \theta_0 x)$ the typical Hopfield energy employed in EP and CHL yields dynamics $s = f(\frac{1}{2}(W + W^\top)s + \theta_0 x)$. So in CHL and EP only the symmetric part of the weight matrix is used. In the setting we are considering here both the symmetric and skew-symmetric part takes part in the dynamics.

With this choice of energy equation 23 becomes

$$\min_{\{W, \theta_0\}} \min_{s^+} \max_{s^-} \tfrac{1}{2}\ell(s^+, y) + \tfrac{1}{2}\ell(s^-, y) + \tfrac{1}{\beta}\left(G(s^+) - G(s^-) - (s^+ - s^-)(W\bar{s} + \theta_0 x)\right). \tag{25}$$

We rewrite the term $(s^+ - s^-)W\bar{s}$ as

$$(s^+ - s^-)W\bar{s} = \tfrac{1}{4}\Big(s^{+\top}(W + W^\top)s^+ + s^{+\top}(W - W^\top)s^- \tag{26}$$
$$- s^{-\top}(W + W^\top)s^- - s^{-\top}(W - W^\top)s^+\Big),$$

which allows us to rewrite the entire objective as

$$\min_{\{W, \theta_0\}} \min_{s^+} \max_{s^-} \tfrac{1}{2}\ell(s^+, y) + \tfrac{1}{2}\ell(s^-, y) \tag{27}$$
$$+ \tfrac{1}{\beta}\Bigg(G(s^+) - G(s^-) - (s^+ - s^-)^\top \theta_0 x$$
$$- \tfrac{1}{4}\begin{pmatrix} s^+ \\ s^- \end{pmatrix}^\top \underbrace{\begin{pmatrix} W + W^\top & W - W^\top \\ -W + W^\top & -W - W^\top \end{pmatrix}}_{\mathcal{W}(W)} \begin{pmatrix} s^+ \\ s^- \end{pmatrix}\Bigg).$$

There are four cases of primary interest here.

- If we do not impose any structure upon $W$ (beyond not allowing self-connections), then equation 27 simply gives us a principled way to train models with arbitrary connectivity. This is significant as CHL and EP rely on symmetric connectivity (see remark A.1) and DP relies on lower triangular (feedforward) connectivity. The drawback is that at training time neurons propagate errors/differences $\tfrac{1}{2}(s^+ - s^-)$ in both bottom-up and top-down direction and the mean $\tfrac{1}{2}(s^+ + s^-)$) in both directions as well. This is a more complicated type of multiplexing than the kind used in DP (where errors flow from top to bottom and means flow from bottom to top).

- Restricting $W$ to be lower triangular. In this setting $W$ transports the mean of activities, $\tfrac{1}{2}(s^+ + s^-)$, forward and $W^\top$ transports the difference backwards $\tfrac{1}{2}(s^+ - s^-)$. This is equivalent to dual propagation in a feedforward model.

- If $W$ is symmetric, then $W - W^\top = 0$, so two blocks of $\mathcal{W}$ vanish, and $s^+$ and $s^-$ decouple, permitting us to infer them sequentially. This is equivalent to equilibrium propagation in a Hopfield model.

- If $W$ is skew-symmetric, then $W + W^\top = 0$, so two blocks of $\mathcal{W}$ vanish. Only the blocks governing interactions between $s^+$ and $s^-$ survive. In general this will require inferring $s^+$ and $s^-$ simultaneously. However, in the case of a layered network (achieved by introducing a particular block sparsity into $W$), then the training dynamics decouple into one minmax problem over odd layers of $s^+$ and even layers of $s^-$ and another minmax problem over even layers of $s^+$ and odd layers of $s^-$.

The three structured weight matrices (lower triangular, symmetric and skew-symmetric) and any smooth combinations thereof can be expressed as $W_\lambda = \theta + (2\lambda - 1)\theta^\top$, where $\theta$ is a lower triangular matrix. The case $\lambda = 1$ yields a symmetric Hopfield model, the case $\lambda = 1/2$ yields a feedforward model and the case $\lambda = 0$ yields a skew-symmetric Hopfield model. Written in terms of $W_\lambda$ equation 27 becomes

$$\min_{\{\theta,\theta_0\}} \min_{s^+} \max_{s^-} \tfrac{1}{2}\ell(s^+, y) + \tfrac{1}{2}\ell(s^-, y) \tag{28}$$

$$+ \tfrac{1}{\beta}\left( G(s^+) - G(s^-) - (s^+ - s^-)^\top \theta_0 x \right.$$

$$\left. - \tfrac{1}{4}\begin{pmatrix} s^+ \\ s^- \end{pmatrix}^\top \begin{pmatrix} \lambda(\theta + \theta^\top) & \bar{\lambda}(\theta - \theta^\top) \\ \bar{\lambda}(-\theta + \theta^\top) & \lambda(-\theta - \theta^\top) \end{pmatrix} \begin{pmatrix} s^+ \\ s^- \end{pmatrix} \right).$$

This formulation is less general than equation 27, but encompasses the three hardware friendly cases. Furthermore, the ability to smoothly interpolate between these models may open the door to efficient pretraining of (skew-) symmetric Hopfield models (by defining an appropriate schedule for $\lambda$).

## B  Mirror descent/ascent dynamics

Here we show that inference of states in the dyadic models can be viewed as mirror descent/ascent (a generalization of the projected gradient method) [35]. In projected gradient descent we aim to take a negative gradient step on an objective $\mathcal{L}$ with respect to a variable $s$ subject to the constraint that the the updated value $s_{t+1}$ is inside a chosen domain $C$ [36] (note that we here use subscripts to denote timesteps rather than layers). This can be formulated as the following problem

$$s_{t+1} = \arg\min_{s \in C} \left\{ s^\top \nabla_s \mathcal{L} + \frac{1}{\eta}||s - s_t||_2^2 \right\}. \tag{29}$$

This yields the updates $s_{t+1} = \pi_C(s_t - \eta\nabla_s E)$, where $\pi_C$ is a projection onto the domain $C$.

In mirror descent the squared Euclidean distance penalizer in equation 29 is replaced with with a Bregman divergence. Bregman divergences are non-negative, but not necessarily symmetric. Familiar examples of Bregman divergences include the squared Euclidean distance, the KL divergence and the Mahalanobis distance. The Bregman divergence between $s$ and $y$ induced by a convex function $G$ is equal to $G(s)$ minus its first order Taylor expansion at $y$ evaluated at $s$.

$$B_G(s, y) = G(s) - \left( G(y) + (s - y)^\top \nabla G(y) \right) \tag{30}$$

Mirror descent with an appropriately chosen Bregman divergence can yield better convergence rates and in some cases turn constrained optimization problems into unconstrained optimization problems [37]. This has recently been used to train quantized neural networks [38].

**Applying mirror descent to the Hopfield setting**  We restrict our analysis to the setting where $G$ is chosen such that it satisfies $\nabla G(u) = f^{-1}(u)$, where f is a (typically nonlinear) activation function. We use as objective $\beta\mathcal{L}_\lambda$ (scaling eq. 4 simplifies things). Using a Bregman divergence $B_G$ induced by $G$ we can replace equation 29 with the following:

$$s_{t+1}^+ = \arg\min_{s^+ \in C} \left\{ s^{+\top} \nabla_{s_t^+} \beta\mathcal{L}_\lambda + \frac{1}{\eta}B_G(s^+, s_t^+) \right\}. \tag{31}$$

$$s_{t+1}^- = \arg\max_{s^- \in C} \left\{ s^{-\top} \nabla_{s_t^-} \beta\mathcal{L}_\lambda - \frac{1}{\eta}B_G(s^-, s_t^-) \right\}. \tag{32}$$

Next we solve for first order optimality with respect to $s$ using the definition of $B_G$ and the fact that $B_G(s^+, s_t^+)$ is convex with respect to its first argument.

$$0 = \pm\nabla_{s^\pm} \beta\mathcal{L}_\lambda + \tfrac{1}{\eta}(\nabla G(s_{t+1}^\pm) - \nabla G(s_t^\pm)) \tag{33}$$

$$\iff$$

$$s_{t+1}^\pm = f(f^{-1}(s_t^\pm) \mp \eta\nabla_{s_t^\pm} \beta\mathcal{L}_\lambda), \tag{34}$$

Using that $\nabla_{s_t^\pm} \beta \mathcal{L}_\lambda = \pm G(s^\pm) \mp \lambda(\theta + \theta^\top)s^\pm \mp \bar{\lambda}(\theta - \theta^\top)s^\mp \mp \theta_0 x \pm \beta \ell'$, and expressing things in terms of the preactivations $a^\pm := f^{-1}(s)$ yields the following dampened fixed point iterations.

$$a_{t+1}^\pm = (1 - \eta)a_t^\pm + \eta\left(\lambda(\theta + \theta^\top)s^\pm + \bar{\lambda}(\theta - \theta^\top)s^\mp + \theta_0 x \mp \beta \ell'\right) \tag{35}$$

Mirror descent allows us to avoid the derivative of the activation function and ensures that the updated states are always feasible (this is tricker to ensure when using projected gradient descent). Non-invertible activation functions such as the sign and relu functions can be seen as limit cases of invertible activation functions such as tanh and leaky-relu.

## C   Adding element-wise contractive non-linearities

We apply implicit differentiation on

$$\nabla G(s) = Ws + x \implies \nabla^2 G(s)\frac{ds}{dx} = W\frac{ds}{dx} + \mathtt{I}.$$

We introduce $\mathtt{D} := \nabla^2 G(s) = (f'(s))^{-1}$, which is a diagonal matrix with elements $\geq 1$ on the diagonal (under the assumption that the activation mapping is 1-Lipschitz, i.e. non-expansive). Therefore

$$\frac{ds}{dx} = (\mathtt{D} - W)^{-1}.$$

On the other hand

$$\mathtt{D} - W = \mathtt{D}^{1/2}(\mathtt{I} - \underbrace{\mathtt{D}^{-1/2}W\mathtt{D}^{-1/2}}_{=:\tilde{W}})\mathtt{D}^{1/2} \tag{36}$$

with $\tilde{W}$ also being skew-symmetric. Consequently

$$\|(\mathtt{D} - W)^{-1}\|_2 = \|\mathtt{D}^{-1/2}(\mathtt{I} - \tilde{W})^{-1}\mathtt{D}^{-1/2}\|_2 \leq \|\mathtt{D}^{-1}\|_2\|(\mathtt{I} - \tilde{W})^{-1}\|_2 \leq 1 \tag{37}$$

since $\mathtt{D}^{-1}$ is a diagonal matrix with entries in $[0, 1]$ and $\|(\mathtt{I} - \tilde{W})^{-1}\|_2 \leq 1$ for any skew-symmetric matrix $\tilde{W}$ (as shown in Section 3, Eq. 16).

