# OpenReview forum: "Dyadic Learning in Recurrent and Feedforward Models"
_NeurIPS.cc/2024/Workshop/MLNCP — MLNCP Oral_

### Official Review · Reviewer_PhnC · 2024-10-04
**The authors investigated closer the role of negative feedback learning and did so using skew symmetric Hopfield networks**

**Rating:** 9
**Confidence:** 2

**Review:**

The paper is well written and follows a clear approach. While not being very familiar with EP and Hopfield networks, the approach sounds convincing to me and the derivation and analysis seems well done.

---

### Official Review · Reviewer_VVoQ · 2024-10-05
**Interesting paper with both theoretical and practical implications**

**Rating:** 9
**Confidence:** 3

**Review:**

This paper develops a particular parametrisation of networks of two-state neurons with the weight-space (including feedback connections) ranging from symmetric to feedforward to skew-symmetric. This enables the authors to develop a generalised saddle-point based learning framework covering Equilibrium Propagation (EP) in symmetric Hopfield models, Dual Propagation in feedforward models, and a novel two-phase learning algorithm in skew-symmetric Hopfield models (SSHM). While the training for SSHM is two-phased just like in EP, there are important differences in what constitutes the two phases of learning. Furthermore, the negative feedback inherent to SSHM also provides them significantly more robustness to large nudging signals during training. They also, in principle, could provide enhanced robustness to adversarial perturbations.

**Strengths:**
- Well-developed novel theoretical framework that covers a range of important local learning approaches.
- SSHMs seem understudied, and this paper highlights potential advantages in exploring them further. I found the larger point about negative feedback having desirable properties in the context of temporal-difference approximations to backprop very interesting.
- Paper is well-written with the motivation and experimental results clearly laid out.

**Potential improvements:**
- The theory is quite dense, and would benefit from a more in-depth explanation of terms and motivations. Given the limited space, this is understandable, but it will be easier to digest in long-form in a full paper.
- More comments on the differences to EP would add to the strength of the paper. What are the implications of having minmax optimisation steps instead of two minimisation steps? What are the potential implications for neuroscience and biological learning? These would all be interesting questions to address or at least provide some comments in the discussion.

---

### Decision · Program_Chairs · 2024-10-10

Accept (Oral)